# eHealth Literacy of Australian Undergraduate Health Profession Students: A Descriptive Study

**DOI:** 10.3390/ijerph191710751

**Published:** 2022-08-29

**Authors:** Carey Ann Mather, Christina Cheng, Tracy Douglas, Gerald Elsworth, Richard Osborne

**Affiliations:** 1Institute of Health Service Management, College of Business and Economics, University of Tasmania, Launceston 7250, Australia; 2Centre for Global Health and Equity, School of Health Sciences, Swinburne University of Technology, Hawthorn 3122, Australia; 3School of Health Sciences, College of Health and Medicine, University of Tasmania, Launceston 7250, Australia

**Keywords:** curriculum, digital, eHealth, eHLQ, health literacy, health profession, student

## Abstract

Rapid growth in digital health technologies has increased demand for eHealth literacy of all stakeholders within health and social care environments. The digital future of health care services requires the next generation of health professionals to be well-prepared to confidently provide high-quality and safe health care. The aim of this study was to explore the eHealth literacy of undergraduate health profession students to inform undergraduate curriculum development to promote work-readiness. A cross-sectional survey was undertaken at an Australian university using the seven-domain eHealth Literacy Questionnaire (eHLQ), with 610 students participating. A one-way Multivariate Analysis of Variance (MANOVA) with follow-up univariate analysis (ANOVA) was used to determine if there were differences in eHLQ scores across 11 sociodemographic variables. Students generally had good knowledge of health (Scale 2); however, they had concerns over the security of online health data (Scale 4). There were also significant differences in age and ownership of digital devices. Students who were younger reported higher scores across all seven eHLQ scales than older students. This research provided an understanding of eHealth literacy of health profession students and revealed sub-groups that have lower eHealth literacy, suggesting that digital health skills should be integrated into university curriculums, especially related to practice-based digital applications with special focus to address privacy and security concerns. Preparation of health profession students so they can efficiently address their own needs, and the needs of others, is recommended to minimise the digital divide within health and social care environments.

## 1. Introduction

Health literacy and eHealth literacy are increasingly recognised as public health issues and determinants of health equity [1,2,3,4,5]. Health literacy strengths and weaknesses across individuals and populations can vary greatly and frequently follow socioeconomic gradients [5,6,7,8]. When people have reduced health literacy, that is, ability to understand, access, appraise, remember, and use health information and health services [4], they tend to have lower engagement in preventive care, increased disease risk factors and poorer health outcomes [1,3,9]. Furthermore, it is well-documented that there are discrepancies between reading skills of intended users and readability of health resources [10,11] with increased demands on community members to access health care through digital means and the move towards health information and services being exclusively provide through digital platforms [12,13].

While the World Health Organisation [14] has recognised that digital health technologies have the potential to strengthen health systems and enable improved health outcomes for individuals and societies [14], access to digital technology and the complexity of the health and social care systems creates barriers [15].The health and social care system in Australia, while includes universal health coverage, creates challenges for healthcare providers and consumers that can lead to poor quality or unsafe care [8,12]. As such, an understanding of eHealth literacy has become an essential research and policy area that requires rapidly development to manage health and social care in the digital era [7,16]

### eHealth Literacy

Exploration and monitoring of consumer eHealth and digital literacy demonstrates the next generation of health profession students will require digital literacy to meet the needs of health and social care consumers [8,17,18]. Current [9,14] governance structures for health technology and digital health at a system, organisational and individual level in Australia remains ad hoc [19,20]. While there is an Australian Digital Health Strategy [12] and Roadmap [21], implementation of health technologies into health and social care environments remains fraught [22,23,24]. Additionally, digital capability of administrative, technical, educational, research and health profession staff need to become ubiquitous [12] within health care environments. Strategic priority 6 of the Australian Digital Health Strategy “A workforce confidently using digital health technologies to deliver health and care” (p. 1) acknowledges that there is a need to upskill the health professional workforce, so that the next generation of health professionals is prepared for the digital future currently being implemented in Australian health and social care environments [12].

To understand and improve digital engagement and the community’s capacity to effectively use eHealth, the field of digital health literacy has emerged. This field began in the web 1.0 era with the emergence of the Lily model, which focused on six technical skills needed to find and use health information from electronic sources [25] and advanced the field through development the eHealth Literacy Scale (eHEALS) questionnaire. Subsequently, Norgaard, Kayser et al. [26] developed the eHealth Literacy Framework (eHLF), a Web 2.0 era tool developed from extensive community consultations [26,27,28,29] using a grounded validity-driven approach to instrument development [30,31]. This model provides a framework to understand individual skills, system requirements, and the interactions between these two. The eHLF formed the basis of a grounded theoretically informed model for measuring eHealth literacy across seven domains:Using technology to process health information;Understanding of health concepts and language;Ability to actively engage with digital services;Feel safe and in control;Motivated to engage with digital services;Access to digital services that work; andDigital services that suit individual needs [26].

There are limited studies investigating the eHealth literacy of undergraduate health profession students. The eHEALS was applied in several studies. A study of pharmacy students in Canada and showed students had a lack of awareness of online health information and ability to use the information to make decisions [32]. A study by Dashti et al. [33] concluded that the eHealth literacy level of medical and health sciences university students in Iran was low while nursing students in Nepal and Sri Lanka were also found to have low eHealth literacy [34,35]. Two Korean studies online health information and determining the quality of such information [36,37]. A recent Danish study using the eHLQ found that the eHealth literacy level of graduate nursing students had a higher eHealth literacy than undergraduate students [38].

Given the rapid transition to digital health platforms, there is a need to understand and integrate eHealth literacy competencies into health profession education [36,37,38,39]. The implementation of digital health technologies into healthcare environments provides impetus to further strengthen health profession curriculums to support undergraduate students to be eHealth literate for the developing digital health environment [12,40]. It is imperative that emerging health profession students enter the field with eHealth literacy competencies [32], so that they can confidently provide high quality and safe health care. Given the limited literature in this area, this study aimed to explore the eHealth literacy of undergraduate health profession students enrolled at an Australian University.

The specific research questions addressed were

What are the eHealth literacy strengths and challenges of undergraduate health profession students as determined by the eHLQ?What sociodemographic factors are associated with the eHealth literacy of undergraduate health profession students?What are the implications for curriculum development with respect to eHealth literacy of undergraduate health profession students?

## 2. Materials and Methods

This explorative descriptive survey was conducted among a convenience sample of undergraduate health profession students enrolled in the College of Health and Medicine at the University of Tasmania. The eHealth Literacy Questionnaire (eHLQ) was administered online using Qualtrics (2018) between July and September 2018. Email invitations were sent to all undergraduate health profession students (9693 students enrolled in a range of different health professions including, medicine, nursing, pharmacy, psychology, paramedicine and a range of health science courses such as dementia care, medical laboratory and radiation science), with one email reminder sent on 9 August 2018. A participant Information and Consent Form was attached with the invitation and return of the survey implied consent. Prior to data collection, ethics approvals were obtained from the Deakin University Human Research Ethics Committee (HEAG-H 146_2017) and the Human Research Ethics Committee (Tasmania) Network (H0016987).

### 2.1. The eHealth Literacy Questionnaire (eHLQ)

The eHLQ, a self-report measure, was used to measure eHealth literacy. The tool, developed based on the eHLF, consists of 35 items with seven scales corresponding to the seven dimensions of the eHLF. Each scale has four to six items. Response options are provided on a 4-point ordinal scale of strongly disagree (1), disagree (2), agree (3) and strongly agree (4).

Initial evidence of reliability and construct validity of the eHLQ obtained during the development study provided evidence that the scale scores derived from the Danish version of the eHLQ would yield appropriate and useful inferences about the eHealth literacy of the participating respondents [31]. Subsequent extensive studies in Australia using the English version in a large primary care sample showed that the tool has robust psychometric properties [28,41]. A further study in Taiwan also provided good evidence of reliability, content and construct validity from data obtained from the urban and regional hospital settings [27].

### 2.2. Data Collection

The survey included the collection of sociodemographic data as the University of Tasmania provides a wide range of opportunities for individuals to study at a tertiary level. Age (early adult, young adult, middle-aged adult, older adult), sex (male, female), socioeconomic status (classified by the Index of Relative Socio-economic Disadvantage (IRSD) Decile, an index about the economic and social conditions of people living within a certain area, see Table 1 footnote), language spoke at home (English, other), education level (secondary school or below, Certificate or Diploma, University or above), private health insurance status (yes, no), health status (presence or absence of longstanding illness), perceived health status (Excellent to good, fair to poor), ownership of digital device, use of digital communication platform, and monitoring health using health technologies such as health apps (yes, no) [42]. Digital devices included computer or laptop, mobile phone, tablet and other. Digital communication platforms included email, text message, Facebook, Twitter, Instagram, Snapchat, WhatsApp or WeChat, blogging, forum, or chat room and other. Data about whether participants looked for information online (yes, no) were also collected. However, this variable was excluded from analysis as only 0.8% (5 out 610) did not look for online information. The survey was designed to be short and anonymous, and no information about course enrolment was collected.

### 2.3. Statistical Analysis

Data were analysed using IBM SPSS Version 25.0 (I.B.M Corporation, Armonk, NY, USA) [43]. The seven eHLQ scores were calculated by summing the item scores and dividing by the number of items in the corresponding scale. Scale scores for an individual respondent were included if there were fewer than two missing item scores in scales with four to five items and fewer than three missing item scores in the scale with six items.

As the seven eHLQ scales are all constructs of eHealth literacy and are not conceptually independent, a one-way Multivariate Analysis of Variance (MANOVA) was used to analyse the mean differences between groups across the 11 sociodemographic variables described [44,45]. An advantage of using MANOVA instead of a series of analyses of variance (ANOVA) is the protection against inflated Type 1 error due to multiple tests of likely correlated dependent variables [46]. There is no consensus on the sample size required for MANOVA analysis. While the sample size is affected by many factors, there are simple recommendations such as a minimum of 20 observations per cell [47] or the size of the smallest group ranging between six to ten times of the number of dependent variables [44]. Hence, the final sample size of 610 was considered adequate (Table 1).

As there is very little research on the eHealth literacy of health profession students, no hypotheses were set for this study. The Pillai’s Trace statistics were reported as unequal samples sizes were present among some of the independent variables and Pillai’s Trace was robust to violations of MANOVA assumptions [46]. A *p*-value of <0.05 was assumed statistically significant. Effect sizes were reported using Eta Squared (η^2^) as independent variables were evaluated individually using one-way MANOVA. Eta squared measures the proportion of the variation in a set of dependent variables that is associated with membership of different groups of an independent variable [48,49] (Levine and Hullett 2002, Richardson 2011). Cohen [50] classifies η^2^ of 0.0099 as a small effect size, i.e., about 1% of variance in the dependent variable is explained by the independent variable, 0.0588 (approximately 6%) being a medium effect and 0.1379 (approximately 14%) considered a large effect size.

If a significant MANOVA difference was found for an individual sociodemographic variable, follow-up univariate ANOVA tests were conducted and further post hoc pairwise comparison tests were undertaken for variables with three or more groups. To determine the relationship between the independent variable and each of the eHealth literacy variables, univariate ANOVAs were considered adequate. For the univariate ANOVAs, a *p*-value of <0.05 was set for statistical significance. The practice of adjusting *p*-value to reduce Type 1 error was considered unnecessary as Huberly and Morris [45] argued that an adequate sample size was sufficient for the protection of Type 1 error. Effect size for each univariate relationship using η^2^ was also reported. For post hoc comparison tests, Tukey Honestly Significant Difference (HSD) test was used. This test was designed to control for Type 1 errors and was applicable to unequal sample sizes among groups [51]. A *p*-value of <0.05 was also considered statistically significant.

## 3. Results

A total of 9693 invitations were sent and 781 were completed, providing a response rate of 8.1%. Of the 781 returned survey, 171 were incomplete and were excluded from analysis, leading to the total sample size of 610 participants.

### 3.1. Participant Characteristics

Sociodemographic characteristics of the sample are shown in Table 1. The age of participants ranged from 19 to 90 with mean (SD) of 44.7 (16.2) (median = 48.0 and mode = 20). The sample was predominately female (80.6%) and most spoke English at home (82.0%). More than half of participants (58.8%) had private health insurance, 52.6% of the sample did not have any longstanding illness and 86.7% of participants perceived their health as good to excellent. On the use of digital technologies, 60.0% owned more than two digital devices, 75.3% engaged with three to five digital communication platforms and 53.2% used digital technologies to monitor health. Ownership and use of digital device was very high. Almost all the sample (99.7%) owned a computer or laptop and a mobile phone (99.0%), while 99.7% used email and 97.9% used text messaging for communication. Facebook was the most used social media (83.6%), followed by Instagram (40.2%).

### 3.2. eHealth Literacy and Sociodemographic Factors

Mean (SD) scores on the seven eHLQ scales are presented in Table 2 (Distribution of scores for each scale is illustrated in Appendix A). The scores ranged from 2.41 to 3.12 (score range: 1 (lowest) to 4 (highest)), with a score below 2.5 indicating people tend to disagree on average with the questionnaire items that constitute the scale. ‘2. Understanding of health concepts and language’ had a mean score of 3.12 (0.45) indicating that participants generally agreed that they had good knowledge of health. However, they might have concerns over the security of their online healthcare data (Scale 4: 2.54 (0.61)). Digital health services may not always be available (Scale 6: 2.48 (0.48)) and may not match their skills or needs (Scale 7: 2.41 (0.56)).

### 3.3. Effects of Sociodemographic Factors on the Combination of the Seven eHLQ Scales

Before conducting the MANOVA, Pearson correlations between the seven dependent variables were tested for the MANOVA assumption of multicollinearity, i.e., they are not highly correlated (0.90 and above in a correlation matrix) which can cause statistical problems [46]. The results showed that a pattern of correlation was observed but none of the correlations exceeded 0.90 indicating the assumption was met (Appendix B).

The MANOVA results identified group differences for eight variables including age, sex, education, language at home, private health insurance, ownership of digital device, use of digital communication platform and monitored health digitally, while no differences were found for socioeconomic status, longstanding illness, and perceived health status. See Table 3.

While significant effects were found for eight variables, the effect sizes were generally small. Only monitored health digitally obtained an effect size of 0.17, indicating that 17.0% of the variance in the combined eHLQ scores was accounted for by monitored health digitally and could be classified as large effect. Medium effect sizes were observed for age (*η*^2^ = 0.06) and ownership of digital devices (*η*^2^ = 0.07).

### 3.4. Effects of Sociodemographic Factors on Individual eHLQ Scales

Following the results of MANOVA, a series of ANOVAs were conducted to identify the specific associations between the eight sociodemographic variables and the individual eHLQ scales that showed significant differences in the multivariate test. Levene’s F tests results were examined to determine if the homogeneity of variance assumption was met. All tests were not statistically significant (*p* > 0.05) except ‘4. Feel safe and in control’ for age (*p* = 0.02), and ‘3. Ability to actively engage with digital services’ (*p* = 0.02) and ‘5. Motivated to engage with digital services’ for monitored health digitally (*p* = 0.01) for monitored health digitally. As the standard deviations of the highest groups divided by the standard deviations of the lowest groups were less than four, ANOVA was considered robust in these cases [52] and the assumption was considered satisfied.

Table 4 presents the univariate F test for the eight independent variables for each of the seven eHLQ scales and the mean score of each group within each independent variable. Results of pairwise comparison tests for variables with three or more groups, i.e., age, education, and use of digital platform, are presented in Appendix C. No large effect size was observed for group differences across any of the eight sociodemographic variables. Medium effect sizes were found in age and monitored health digitally for some scales. For age, Scales ‘2. Understanding of health concepts and language’, ‘3. Ability to actively engage with digital services’, ‘4. Feel safe and in control’ and ‘7. Digital services that suit individual needs’ recorded η^2^ ranging from 0.06 to 0.10. For monitored health digitally, medium effect sizes were obtained for five scales including Scales ‘1. Using technology to process health information’, ‘2. Understanding of health concepts and language’, ‘3. Ability to actively engage with digital services’, ‘5. Motivated to engage with digital services’ and ‘7. Digital services that suit individual needs’ with η^2^ ranging from 0.08–0.13.

From the perspective of the dependent variables, Scale ‘2. Understanding of health concepts and language’ was significantly associated with all sociodemographic factors except language at home. Three scales including ‘3. Ability to actively engage with digital services’, ‘5. Motivated to engage with digital services’ and ‘7. Digital services that suit individual needs’ were associated with six independent variables. ‘1. Using technology to process health information’ was related only to three variables: age, use of digital platform and monitored health digitally.

To gain further insights into the group differences, group mean scores were examined for variables with two groups. For scales that showed group differences, males scored higher than females, and the same applied to participants with more devices versus those with less devices, as well as participants who monitored health digitally versus those who did not. Pairwise comparison tests (Appendix C) further showed that younger age groups generally reported significantly higher scores in all scales. Participants with secondary or below education reported higher scores than those with higher education for Scales ‘4. Feel safe and in control’, ‘5. Motivated to engage with digital services’, ‘6. Access to digital services that work’ and ‘7. Digital services that suit individual needs’. On the use of digital communication platform, low users reported lower scores comparing to medium or high users in all seven scales.

## 4. Discussion

### 4.1. Principal Findings

This study explored eHealth literacy of undergraduate health profession students enrolled in an Australian university. The purpose of this research was to understand the eHealth literacy needs of health profession students to inform curriculum development to prepare students for the digitally enhanced healthcare environment that has now arrived. This study found that participants believed they generally had strong health knowledge and do use technology for health. However, they indicated some concerns over online security while digital health services that met their skills or needs might not always be available. Participants who were younger, owned more digital devices, used more digital communication platforms, and monitored their health digitally, reported higher eHealth literacy scores than others who participated in this study.

### 4.2. eHealth Literacy

Overall, Australian undergraduate students tended to report perceived knowledge and were comfortable using technology for health. However, they expressed concerns regarding security of online health data (‘4 Feel safe and in control’: 2.54). This finding may relate to health data security breaches reported at local [53] and national [54] levels in Australia. Additionally, Australian health profession students undertaking practice-based work integrated learning are not provided with passwords to access healthcare organisation intranets or healthcare-related data as they are not employees. Students undertaking work integrated learning rely on using their host supervisor’s passwords which models poor cyber safety [55] and may reinforce student perceptions about privacy and security of data.

Respondents also indicated they may not have satisfactory access to digital health services (‘6. Access to digital services that work’: 2.48) and services may not meet their abilities or needs (‘7. Digital services that suit individual needs’: 2.41). This finding may be related to respondent perceptions of the Australian electronic health record (EHR). The roll out of the Australian EHR has been problematic due to a range of factors including lack of health literacy [8], lead time, educational preparation of health professionals and security and privacy issues [22,23,24]. Although health professionals have the capacity to upload information to EHR, the level of completeness of records varies [22]. A lack of training in the use of the EHR and lack of trust in security of data by health professionals and the population has impeded the quality of information available to health professionals regarding recipient of care records [22,23,24]. Exposure to electronic systems in clinical settings may be limited by the curriculum and the capacity of health professionals to access digital services and relevant technologies [38].

### 4.3. Sociodemographic Factors

Among all the sociodemographic factors, only ‘monitored health digitally’ (Table 1) was positively related to eHealth literacy with a large effect, while ‘owning more digital devices’ (Table 1) was linked to higher eHealth literacy with a medium effect. These findings are not surprising as participants who monitored their health digitally were likely to be comfortable using technologies for health which will contribute to higher eHealth literacy scores across the seven scales. Participants who owned more digital devices would also have more opportunity to use these devices for health, in addition to using them for other purposes. While monitoring health digitally may be beneficial, how they do it may be cause for concern as wearables, apps and online tools are not governed by standards or guidelines [56]. This lack of direction impedes the capacity of current health professionals or students in assessing the credibility or efficacy of the burgeoning range of wearables, apps or online digital health tools [56]. Further studies into the prescription of these products are warranted, to ensure safe, effective, and appropriate use. Concurrently, educators need to be abreast of this evolving sub-field within digital health to ensure health profession students are equipped with the critical thinking and decision-making skills regarding digital monitoring of students’ own health and of the health of others in their care.

Being older was associated with a lower eHealth literacy score with a medium combined effect. This finding may reflect the use of digital platforms by different age groups in the cohort, echoing the findings of research by Holt et al. [38], which also found significant effects of age among Danish nursing students. However, age was not found to be a factor affecting eHealth literacy in the study involving Korean nursing students [37], possibly because the Korean cohort was younger than the cohort in this study.

Education only had a modest relationship on eHealth literacy. Nevertheless, participants with secondary or below education reported higher scores in Scales ‘4. Feel safe and in control’, ‘5. Motivated to engage with digital services’, ‘6. Access to digital services that work’ and ‘7. Digital services that suit individual needs’ than higher educated groups. Implementation of health literacy programs at secondary education level [57,58] and concurrent decrease in costs of digital technology may have contributed to a younger generation who are more likely to be comfortable, motivated as well as having good access to use digital technology for health [59,60,61]. Similarly, this study found younger people reported significantly higher eHealth literacy scores across all seven scales of the eHLQ.

### 4.4. Implications for Curriculum Development

Findings of this study indicated students have embraced using digital devices and platforms for health but expressed concerns for security as well as good access to digital services that suit individual needs. Learners need to develop a digitally professional approach to healthcare delivery during their practice-based or virtual work integrated learning experiences [62]. Students need to develop capability in digital professionalism, which is a component of professional identity formation prior to undertaking practice-based work integrated learning [20,55,62]. The curriculum needs to include specific information related to privacy and security issues from an individual, organisation and systems approach. While in the Australian context, the Australian digital Health Agency National Digital Health Strategy [12] and Roadmap [21] provides direction regarding eHealth literacy for the higher education sector, translation at a local level has been problematic. Cost, equity of access, educator confidence or organisational readiness [20] has hindered implementation, which needs to be addressed.

According to Holt et al. [38], current curricula and study activities in nursing curriculum in Denmark are appropriate to support eHealth literacy at a satisfactory level although additional learning opportunities involving health, digital and eHealth literacy could improve graduate outcomes. The authors found that eHealth literacy relating to personal knowledge and skills was higher among graduate-level, than entry-level students indicating that curriculum and experience as a nursing student had a positive impact on eHealth literacy. The study also found that graduate nursing students demonstrated a more critical attitude towards digital services that suit digital needs which was assumed to be related to their experiences of direct contact with digital services during their clinical training as a health professional. An integration of eHealth literacy using practice-based applications should be visible early in the undergraduate curriculum [63]. This integration should involve eHealth information seeking activities which are embedded throughout the curriculum to enhance the proficiency of graduate health education students to use digital resources to locate and evaluate eHealth information.

Health literacy in Tasmania health profession students has been previously identified as needing improvement supporting the opportunity to embed modules of health literacy in the current curriculum [6]. Table 1 shows the socioeconomic disadvantage of Tasmanian health profession students which supports the findings of previous research, Similarly, there is a need to embed eHealth literacy into the curriculum to enhance the digital capabilities of health profession graduates. As identified by Kim and Jeon [37], a multidimensional approach is required to reform nursing education to improve eHealth literacy and ensure that students receive eHealth literacy in an integrated curriculum throughout their university studies. Students enter health courses from a diverse range of backgrounds and can therefore be expected to have differing levels of health literacy and eHealth literacy. This finding was evident in this study as being an older student was associated with lower levels of eHealth literacy. A self-assessment survey of current digital literacy level of students at the commencement of their higher education studies in health would enable students to be identified regarding individual needs to enhance their digital literacy skills. Embedding a framework to initially determine current eHealth literacy status [26] is also warranted. Identified students could then be directed towards appropriate modules of activity to improve their skills. The course curriculum could contain a module in eHealth literacy to be completed prior to practice-based work integrated learning to ensure that students are well-placed to address their eHealth literacy needs prior to [26] attending work integrated learning. Within the placement experience of their course, a further module on how to utilise digital health information effectively could be completed by students and incorporated into their professional portfolio. Development of digital professionalism could become core content across a variety of health courses in the university setting. Once an eHealth literacy curriculum is embedded, further research within health disciplines, wider student community and population to evaluate change will determine any eHealth literacy improvements, as a consequence of implementation, will be necessary.

Strengthening curriculum to further support students to develop capability in eHealth literacy and digital professionalism includes learning and understanding about safe and appropriate access and use of digital technologies. Developing a strong understanding of privacy and security issues is a component of developing professional identity as a health professional. It is mandatory that health profession students understand their scope of practice in relation to using digital technology when undertaking work integrated learning [64]. Most higher education health profession courses require students to undertake practice-based or virtual work integrated learning, where they augment their on-campus learning through simulated or ‘real world’ activities, which promotes students to develop proficiency in eHealth literacy. It is important all undergraduate health profession students regardless of age gain a base-line level of eHealth literacy and digitally professional behaviour prior to graduation [19]. The curriculum needs to support and enable all students enrolled in health profession courses at the University of Tasmania, and other Australian universities, to become educationally prepared and work ready in both eHealth literacy and digital professionalism.

### 4.5. Limitations and Future Directions

The low response rate (8.06%) and self-selection of respondents within this cohort may reduce generalisability of the findings. Compared with non-respondents, it is possible that respondents had higher eHealth literacy and may have also been more likely to have had a lived experience of health issues and therefore were more interested in taking part in the survey. Fatigue of undergraduate students, and their lack of recognition of the relevance of this survey to their future career may have contributed to the low response rate [65]. eHealth literacy is an emerging field and the predominance of female participants (80.6%) reduced gaining a representative sample of respondents. However, gender bias is a common phenomenon among health profession student studies. Holt and colleagues [36] and Park and Min [32] also had 92% and 87% of participants being female, respectively. An important future direction of research is to obtain representative samples. Achievement of this objective may be through inclusion of incentives to encourage participation, or incorporation of eHealth literacy surveys in government sponsored national health surveys.

Since the COVID-19 pandemic there has been a pivot to increased online or blended learning which suggests future work needs to obtain representative samples, collect sufficient details of health profession courses enrolled in, which would provide more precise data to prioritise educational preparation and support for students.

## 5. Conclusions

This study provides a baseline understanding of the eHealth literacy of health profession students that can be utilised to further improve curriculum design to ensure the next generation of health profession students are more educationally prepared and work-ready at graduation. The integration of eHealth literacy should be prominent in the undergraduate curriculum using practice-based digital applications. Digital professionalism should become core content across a variety of health courses. The diverse eHealth literacy needs of health profession students should also be addressed by directing students to the appropriate module of activities to improve their knowledge and skills and develop their capability in eHealth literacy. Inclusion of eHealth literacy within health profession curriculums will educationally prepare the next generation of health professionals for the digitally enhanced healthcare environment that has now arrived.

## Figures and Tables

**Table 1 ijerph-19-10751-t001:** Sociodemographic characteristics of participants (*n* = 610).

Characteristics	*n* (%)	Missing Data (*n*)
Age (Range 19–90, mean (SD) 44.7 (16.2))	597	13
Early adult (19–25)	129 (21.1)	
Young adult (26–40)	99 (16.2)	
Middle-aged adult (41–55)	186 (30.5)	
Older adult (56–90)	183 (30.0)	
Sex	605	5
Female	487 (79.8)	
Male	118 (19.3)	
Education	610	0
Secondary school or below	138 (22.6)	
Certificate or Diploma	235 (38.5)	
University or above	237 (38.9)	
Language at home	608	2
English	497 (81.5)	
Other	111 (18.2)	
Socioeconomic status (SES) *	574	36
IRSD 1–2	135 (22.1)	
IRSD 3–4	99 (16.2)	
IRSD 5–6	77 (12.6)	
IRSD 7–8	145 (23.8)	
IRSD 9–10	118 (19.3)	
Longstanding illness	608	2
Yes	287 (47.0)	
No	321 (52.6)	
Perceived health status	610	0
Good to Excellent	529 (86.7)	
Fair to Poor	81 (13.3)	
Private health insurance	610	0
Yes	358 (58.7)	
No	252 (41.3)	
Ownership of digital device	610	0
Less devices (owned 1–2 devices)	245 (40.2)	
More devices (owned 3–4 devices)	365 (59.8)	
Owned computer/laptop	608 (99.7)	
Owned mobile phone or smartphone	604 (99.0)	
Owned tablet	366 (60.0)	
Owned other device	32 (5.2)	
Use of digital communication platform	610	
Low use (used 1–2 platforms)	54 (8.9)	
Medium user (used 3–5 platforms)	458 (75.1)	
High user (used 6–9 platforms)	98 (16.1)	
Used email	608 (99.7)	
Used text message	597 (97.9)	
Used Facebook	510 (83.6)	
Used Twitter	69 (11.3)	
Used Instagram	245 (40.2)	
Used Snapchat	181 (29.7)	
Used WhatsApp/WeChat	161 (26.4)	
Used blogging	25 (4.1)	
Used forum/chat room	138 (22.6)	
Used other communication platform	16 (2.6)	
Look for online information	610	0
Yes	605 (99.2)	
No	5 (0.8)	
Monitored health digitally	610	0
Yes	325 (53.3)	
No	285 (46.7)	

* SES is classified by IRSD10–The Index of Relative Socio-economic Disadvantage Decile 2016, ranking within Australia. This index is based on information provided by the Australian Bureau Statistics (2018). Postcodes are divided into 10 ranks with higher number indicating more advantaged suburbs.

**Table 2 ijerph-19-10751-t002:** eHLQ scores for overall sample (score range: 1 (lowest) to 4 (highest).

	Mean (SD), [95% CI] *	Missing Values (n)
1. Using technology to process health information	2.82 (0.48) [2.78–2.85]	0
2. Understanding of health concepts and language	3.12 (0.46) [3.0–3.16]	0
3. Ability to actively engage with digital services	2.95 (0.55) [2.90–2.99]	0
4. Feel safe and in control	2.54 (0.62) [2.49–2.59]	0
5. Motivated to engage with digital services	2.69 (0.52) [2.65–2.73]	0
6. Access to digital services that work	2.49 (0.48) [2.45–2.52]	1
7. Digital services that suit individual needs	2.41 (0.56) [2.36–2.45]	3

* SD, standard deviation, CI, confidence interval.

**Table 3 ijerph-19-10751-t003:** Relationship between sociodemographic variables and the combination of the seven eHLQ scales.

Variable	Pillai’s Trace	*F*	*df*	*Error df*	*p*	*η*^2^ *
Age	**0.19**	5.61	21	1758	<0.001	0.06
Sex	**0.03**	2.29	7	594	0.03	0.03
Education	**0.09**	3.94	14	1198	<0.001	0.04
Language at home	**0.04**	3.35	7	597	0.00	0.04
SES	0.06	1.12	28	2252	0.30	0.01
Longstanding illness	0.01	1.05	7	597	0.40	0.01
Perceived health status	0.02	1.42	7	599	0.19	0.02
Private health insurance	**0.03**	2.85	7	599	0.01	0.03
Ownership of digital device	**0.07**	6.34	7	599	<0.001	0.07
Use of digital communication platform	**0.08**	3.64	14	1198	<0.001	0.04
Monitored health digitally	**0.17**	17.19	7	599	<0.001	0.17

Results in bold indicate significant with *p* < 0.05 * Small effect size *η*^2^ = 0.0099–0.0587; Medium effect size *η*^2^ = 0.0588–0.1379; Large effect size *η*^2^ ≥ 0.1379.

**Table 4 ijerph-19-10751-t004:** Relationship between sociodemographic characteristics and individual eHealth Literacy Questionnaire (eHLQ) scales.

		1. Using Tech	2. Health Concepts	3. Engage	4. Feel Safe	5. Motive	6. Access	7. Suit Needs
**Age**								
*F* (3, 593)		**10.46**	**13.18**	**22.07**	**12.82**	**10.90**	**7.37**	**14.06**
*p*		**<0.001**	**<0.001**	**<0.001**	**<0.001**	**<0.001**	**<0.001**	**<0.001**
*η* ^2^		**0.05**	**0.06**	**0.10**	**0.06**	**0.05**	**0.04**	**0.07**
Group	*n*	Mean score (SD) [95% CI]						
Early adult (age 19–25)	129	**2.96 (0.53)** [2.87, 3.05]	**3.31 (0.45)** [3.24, 3.39]	**3.16 (0.47)** [3.07, 3.24]	**2.82 (0.52)** [2.73, 2.91]	**2.84 (0.56)** [2.75, 2.94]	**2.66 (0.47)** [2.57, 2.73]	**2.62 (0.54)** [2.52, 2.71]
Young adult (age 26–40)	99	**2.94 (0.44)** [2.84, 3.02]	**3.17 (0.45)** [3.08, 3.26]	**3.17 (0.49)** [3.07, 3.26]	**2.49 (0.64)** [2.36, 2.62]	**2.83 (0.49)** [2.74, 2.93]	**2.45 (0.49)** [2.35, 2.55]	**2.53 (0.53)** [2.42, 2.63]
Middle-aged adult (age 41–55)	186	**2.77 (0.44)** [2.70, 2.83]	**3.08 (0.42)** [3.02, 3.14]	**2.89 (0.53)** [2.81, 2.96]	**2.53 (0.59)** [2.44, 2.61]	**2.64 (0.49)** [2.57, 2.72]	**2.48 (0.47)** [2.41, 2.55]	**2.38 (0.53)** [2.31, 2.46]
Older adult (age 56–90)	183	**2.71 (0.45)** [2.64, 2.77]	**3.01 (0.44)** [2.95, 3.07]	**2.75 (0.58)** [2.67, 2.83]	**2.40 (0.61)** [2.32, 2.50]	**2.56 (0.50)** [2.49, 2.64]	**2.40 (0.47)** [2.34, 2.47]	**2.24 (0.56)** [2.16, 2.32]
**Sex**								
*F* (1, 603)		2.60	**6.96**	**9.70**	0.31	**5.97**	0.37	0.86
*p*		0.12	**0.01**	**0.00**	0.58	**0.02**	0.54	0.35
*η* ^2^		0.00	**0.01**	**0.02**	0.00	**0.01**	0.00	0.00
Group	*n*	Mean score (SD) [95% CI]						
Male	118	2.88 (0.51) [2.78, 2.97]	**3.22 (0.47)** [3.13, 3.30]	**3.09 (0.53)** [2.99, 3.18]	2.57 (0.65) [2.46, 2.69]	**2.80 (0.52)** [2.70, 2.89]	2.51 (0.52) [2.42, 2.61]	2.45 (0.60) [2.34, 2.56]
Female	487	2.80 (0.47) [2.76, 2.84]	**3.10 (0.45)** [3.06, 3.14]	**2.91 (0.56)** [2.86, 2.96]	2.54 (0.60) [2.48, 2.59]	**2.67 (0.52)** [2.62, 2.71]	2.48 (0.48) [2.44, 2.52]	2.40 (0.55) [2.35, 2.45]
**Education**								
*F* (2, 607)		2.56	**10.85**	**4.73**	**13.29**	**4.35**	**11.00**	**5.35**
*p*		0.08	**<0.001**	**0.01**	**<0.001**	**0.01**	**<0.001**	**0.01**
*η* ^2^		0.01	**0.04**	**0.02**	**0.04**	**0.01**	**0.04**	**0.02**
Group	*n*	Mean score (SD) [95% CI]						
Secondary or below	138	2.90 (0.52) [2.81, 2.98]	**3.24 (0.44)** [3.16, 3.31]	**3.07 (0.49)** [2.98, 3.14]	**2.77 (0.59)** [2.67, 2.87]	**2.81 (0.53)** [2.72, 2.90]	**2.64 (0.50)** [2.56, 2.73]	**2.55 (0.56)** [2.45, 2.64]
TAFE/Diploma	235	2.79 (0.46) [2.73, 2.85]	**3.02 (0.45)** [2.97, 3.08]	**2.89 (0.55)** [2.81, 2.96]	**2.50 (0.60)** [2.42, 2.58]	**2.65 (0.50)** [2.58, 2.71]	**2.48 (0.49)** [2.42, 2.54]	**2.37 (0.57)** [2.30, 2.44]
University or above	237	2.79 (0.47) [2.73, 2.85]	**3.15 (0.46)** [3.09, 3.21]	**2.93 (0.58)** [2.86, 3.01]	**2.44 (0.61)** [2.37, 2.52]	**2.67 (0.52)** [2.60, 2.74]	**2.40 (0.45)** [2.35, 2.46]	**2.37 (0.55)** [2.30, 2.44]
**Language at home**								
*F* (1, 606)		0.17	1.99	0.03	2.70	**3.91**	3.72	**10.37**
*p*		0.68	0.16	0.86	0.10	**0.05**	0.05	**0.00**
*η* ^2^		0.00	0.00	0.00	0.00	**0.01**	0.01	**0.02**
Group	*n*	Mean score (SD) [95% CI]						
English	497	2.81 (0.48) [2.77, 2.85]	3.13 (0.46) [3.09, 3.17]	2.94 (0.57) [2.89, 2.99]	2.52 (0.61) [2.47, 2.57]	**2.67 (0.53)** [2.63, 2.72]	2.47 (0.48) [2.43, 2.51]	**2.37 (0.56)** [2.32, 2.42]
Other	111	2.83 (0.44) [2.75, 2.91]	3.07 (0.41) [2.99, 3.14]	2.95 (0.50) [2.86, 3.05]	2.63 (0.63) [2.51, 2.75]	**2.78 (0.48)** [2.69, 2.87]	2.57 (0.47) [2.48, 2.65]	**2.56 (0.55)** [2.46, 2.67]
**Private health insurance**								
*F* (1, 608)		0.42	**5.19**	0.34	0.12	0.13	**4.92**	**4.06**
*p*		0.52	**0.02**	0.56	0.73	0.72	**0.03**	**0.04**
*η* ^2^		0.00	**0.01**	0.00	0.00	0.00	**0.01**	**0.01**
Group	*n*	Mean score (SD) [95% CI]						
Yes	358	2.80 (0.49) [2.76, 2.86]	**3.15 (0.45)** [3.10, 3.20]	2.96 (0.57) [2.90, 3.01]	2.53 (0.62) [2.47, 2.60]	2.69 (0.53) [2.63, 2.74]	**2.45 (0.47)** [2.40, 2.50]	**2.37 (0.56)** [2.31, 2.43]
No	252	2.83 (0.46) [2.78, 2.89]	**3.07 (0.45)** [3.01, 3.13]	2.93 (0.53) [2.86, 3.00]	2.55 (0.61) [2.47, 2.62]	2.70 (0.51) [2.64, 2.77]	**2.54 (0.50)** [2.48, 2.60]	**2.46 (0.56)** [2.39, 2.53]
**Ownership of digital device**								
*F* (1, 608) *p* *η*^2^		3.15 0.08 0.01	**4.72** **0.03** **0.01**	**27.01** **<0.001** **0.04**	2.04 0.15 0.00	3.49 0.06 0.01	0.32 0.57 0.00	1.34 0.25 0.00
Group	*n*	Mean score (SD) [95% CI]						
Less device (1–2 devices)	245	2.77 (0.45) [2.71, 2.83]	**3.07 (0.47)** [3.01, 3.13]	**2.80 (0.54)** [2.74, 2.87]	2.58 (0.59) [2.51, 2.66]	2.64 (0.50) [2.58, 2.71]	2.50 (0.50) [2.44, 2.56]	2.38 (0.56) [2.31, 2.45]
More device (3–4 devices)	365	2.84 (0.49) [2.79, 2.89]	**3.15 (0.44)** [3.11, 3.20]	**3.04 (0.54)** [2.98, 3.09]	2.51 (0.63) [2.44, 2.57]	2.73 (0.53) [2.67, 2.78]	2.48 (0.48) [2.43, 2.53]	2.43 (0.56) [2.37, 2.49]
**Use of digital communication platform**								
*F* (2, 607) *p* *η*^2^		**9.42** **<0.001** **0.03**	**7.85** **<0.001** **0.03**	**9.60** **<0.001** **0.03**	**7.17** **<0.001** **0.02**	**13.56** **<0.001** **0.04**	**3.14** **0.04** **0.01**	**10.48** **<0.001** **0.03**
Group	*n*	Mean score (SD) [95% CI]						
Low user (1–2 platforms)	54	**2.57 (0.53)** [2.42, 2.71]	**2.89 (0.53)** [2.75, 3.04]	**2.74 (0.61)** [2.58, 2.90]	**2.24 (0.60)** [2.09, 2.41]	**2.37 (0.56)** [2.23, 2.54]	**2.35 (0.54)** [2.22, 2.50]	**2.14 (0.57)** [1.98, 2.29]
Medium user (3–5 platforms)	458	**2.83 (0.46)** [2.78, 2.87]	**3.14 (0.44)** [3.10, 3.18]	**2.93 (0.55)** [2.88, 2.98]	**2.55 (0.61)** [2.50, 2.61]	**2.70 (0.51)** [2.65, 2.75]	**2.49 (0.47)** [2.44, 2.53]	**2.41 (0.56)** [2.36, 2.46]
High user (6–9 platforms)	98	**2.90 (0.87)** [2.81, 3.00]	**3.17 (0.45)** [3.08, 3.26]	**3.13 (0.48)** [3.03, 3.23]	**2.63 (0.60)** [2.51, 2.75]	**2.83 (0.47)** [2.73, 2.92]	**2.56 (0.51)** [2.45, 2.66]	**2.57 (0.52)** [2.46, 2.68]
**Monitored health digitally**								
*F* (1, 608) *p* *η*^2^		**83.08** **<0.001** **0.12**	**57.10** **<0.001** **0.09**	**78.54** **<0.001** **0.11**	**19.82** **<0.001** **0.03**	**87.38** **<0.001** **0.13**	**21.01** **<0.001** **0.03**	**52.31** **<0.001** **0.08**
Group	*n*	Mean score (SD) [95% CI]						
Yes	325	**2.97 (0.44)** [2.92, 3.02]	**3.24 (0.43)** [3.20, 3.29]	**3.12 (0.48)** [3.07, 3.17]	**2.64 (0.59)** [2.58, 2.70]	**2.87 (0.47)** [2.82, 2.92]	**2.57 (0.48)** [2.52, 2.62]	**2.56 (0.52)** [2.50, 2.62]
No	285	**2.64 (0.46)** [2.59, 2.69]	**2.98 (0.44)** [2.93, 3.03]	**2.75 (0.56)** [2.68, 2.80]	**2.42 (0.63)** [2.35, 2.50]	**2.50 (0.51)** [2.44, 2.56]	**2.39 (0.48)** [2.34, 2.45]	**2.24 (0.56)** [2.18, 2.31]

Results in bold have *p*-value of <0.05 for significant differences in means. Small effect size *η*^2^ = 0.0099; Medium effect size *η*^2^ = 0.0588; Large effect size *η*^2^ = 0.1379.

## Data Availability

Data are held by The Centre for Global Health and Equity, School of Health Sciences. Request for data must be made in writing to C.C.

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
