# Peer review of "eHealth Literacy of Australian Undergraduate Health Profession Students: A Descriptive Study"

_ijerph, 2022, doi:10.3390/ijerph191710751_

Round 1
Reviewer 1 Report
Dear Authors
This paper is about a significant issue for health professionals But after a careful review the decision is major revision for the following reasons.
The aim of the study was expressed as “this study aimed to explore the eHealth literacy of undergraduate health profession students enrolled at an Australian University”. This purpose sentence is insufficient to evaluate the text. It would be helpful to add research questions that include dependent and independent variables.
In method section, the fundamental problem is to state the desing of the study as cross-sectional.
Although the design of the study was defined as cross-sectional, which is one of the epidemiological analytical designs, the study does not have cross-sectional design features.
First of all, while the cross-sectional design is expected to be a study that represents the society, it is seen that this study is based on a non-probabilistic sampling method from undegraduate helath professionals in a university, the “e-mail invitation”, a sample selection method that can not be representative of the target population. It is appropriate to express the study design as descriptive rather than cross-sectional.
In the method section of the study, it was not specified how the sample size was determined before the research. The suitability of the number of participants reached after the study was evaluated only for suitibility for the statistical analyses. How many people did the target group consist of? “A total of 9,693 invitations were sent and 781 were completed, providing a response rate 190 of 8.1%”. Is 9693, the number of target groups or just the number of people to whom e-mail is sent?
What were the characteristics of the target group? Only “health professionals with undergraduate education” were used. In the following sections, the expressions of nursing students are used. Was the study group made up of nursing students only? Were there also other healthcare professionals? All this information will be important in interpreting the study results.
In addition, the study group was defined as university undergraduate students, and In the results section, it was stated that the education level of the participants was "Secondary school or below, Certificate or Diploma, University or above"...Education levels of the participants makes it necessary to explain what kind of education this university expects in accepting students at the undergraduate level or how the undergraduate education is defined in this university. Was it different description from the universal criteria for undergraduate level education.
Among the items of the eHLQ scale, pearson correlations were used to decide on some statistics. It is expected that a scale whose reliability has been tested before has integrity and that there is a correlation between its items. It is unnecessary to examine this situation again.
The purpose of the study, the definition of the study design, the number and characteristics of the target population, the characteristics of the study group, and the size of the study group need to be reviewed.
Reviewer 2 Report
Thank you very much for the opportunity to read about an important topic that arises with the increasing rate of eHealth system adoption across the globe.
Though I consider this topic interesting and important for readers, I have some issues and remarks that I would like to share with you:
"A study by Dashti et al [30] concluded that the eHealth literacy level of medical and health sciences university students in Iran was low while nursing students in Nepal and Sri Lanka were found to have low eHealth literacy [31, 32]." Does it mean that they have been both on the low level?
I am not entirely familiar with the Australian education system, however, in my country, it would not be typical to find many 40+ undergraduate level students (the mean age reported in the article is 44.7 years), not even mentioning students from the 56-90 cohort. Is this a typical situation that reflects the distribution across the whole country, or is it something specific for a single institution? In the latter case, I have some doubts about how it can reflect the situation countrywise (leading to some information bias when talking about the results to be useful for the stakeholders responsible for the curriculum design).
Further, I expect that the cohort structure as you present it is also associated with the fact that there is a large group with longstanding illnesses in your sample. I would like to know - referring to some broader national health statistics - whether, indeed, approx. 47 % of all students across the whole of Australia are experiencing some kind of a longstanding illness. In any case, the Discussion section should, in my opinion, at least mention the possibility that the longstanding illness might have been one of the reasons that actually lead the final sample to respond to this questionnaire in the first place.
I am also not entirely sure what the purpose of asking the participants whether they are using digital communication platforms in such a great level of detail is. Is there a difference for example between Twitter or Facebook users from the point of digital/eHealth literacy? Wouldn't it be better to simply count how many students use at least some kind of social networks, or how many of them they are using?
The authors decided to analyze the eHealth literacy using the eHLQ questionnaire which was administered via an on-line platform. This is already expecting a somewhat higher baseline level of digital literacy of the respondents which is an integral part of the eHealth literacy. It is also interesting that the invitation was send via e-mail and the usage of e-mail is actually reported by 99.7 % of respondents. That probably means that those 0.3 % of the participants should have been excluded as well.
Based on the response rate, it is worth asking whether the group that actually completed the questionnaire has some other common (personal) traits that can be further assessed.
eHLQ is a self-reporting questionnaire. It is doubtful to what extent the self-confidence of the participants is actually tested, rather than their actual eHealth literacy. Especially when comparing different age cohorts. I am well aware that the eHLQ is a validated questionnaire and it psychometric validity and reliability has been shown in papers, nevertheless, a performace test would be more beneficial in the case of eHealth literacy assessment. Especially among younger generation which was growing up with smartphones and tablets and thinks that their digital literacy is build up around the entertainment applications and social networks. Nevertheless, saying nothing about their true eHealth literacy. From that point of view, only a question about digital health monitoring usage is giving some more reliable information, as it shows a practical experience gained by each participant. From my perspective, an informal vital sign monitoring, for example for the purpose of individual physical training and consequent work with e.g. the Google fit data, are telling us more about the eHealth literacy than the other domains. After all, the authors are reaching a similar conclusion: "Among all the sociodemographic factors, only ‘monitored health digitally’ was positively related to eHealth literacy with a large effect, while ‘owning more digital devices’ was linked to higher eHealth literacy with a medium effect."
For the presented study, no hypotheses have been designed. That is not an issue per se. However, with the amount of different tests conducted, it gives an impression of looking up for significant results to arise. From that perspective, I think that a descriptive analysis and some simple correlations would be good enough unless looking for a specific answer.
I have strong doubts that based on this study, any substantial (and specific) change or improvement of the curriculum can be implemented. If there are differences among the age cohorts of the students attending the same study programme, it is obvious that they gather their knowledge elsewhere, especially via informal learning. Therefore, I am not confident and convinced that the main purpose described in the paper can be considered as fulfilled. More precise performance testing of the students will need to be exploited to reach this goal and establish a feedback loop between their knowledge and the actual curriculum design.
I can relate that the EHR competences are a major issue. Yet, this is just another example of what should be better assessed using performance testing of the respective users rather than by self-reporting questionnaires. As the subjective perspective of the users might be distorted by their personal opinions.
Last but not least, it is interesting that the data gathered in 2018 are being published now. That raises some questions whether there were some previous attempts to publish elsewhere.
I think that the data can be succesfuly used, if the same group can be reached once again and checked whether there is some substantial change (improvement) after the COVID, as one would expect that the urge to use the eHealth has increased. Therefore, it might be useful to assess that by the proposed eHLQ together with some more specific questions beyond those available in eHLQ, an improvement took place in the monitored group during the pandemic.
For the successful publication, it is important to at least know that the research group reflects the student group across the Australia. Otherwise, the current title is strongly misleading (as it does not offer countrywise baseline) and should be rephrased for example to: eHealth literacy of undergraduate health profession students: a cross sectional study of the University of Tasmania.
Please recheck your affiliations.
Reviewer 3 Report
This study explores the eHealth literacy of undergraduate health profession students to inform undergraduate curriculum development to promote work-readiness. A cross-sectional survey was undertaken using the 7-domain eHealth Literacy Questionnaire (eHLQ). The work concludes that inclusion of eHealth literacy within health profession curriculums will educationally prepare the next generation of health professionals.
The present reviewer finds that this manuscript needs to be improved before considering its publication in this journal. The observations are detailed below:
1. The explanation of the 7 domains on which this research is based is brief and not sufficiently justified, so it is suggested that the authors discuss the application of these domains against other possible scenarios for conducting this study.
2. In the results section, it is notable that the authors do not present other related results from other investigations that discuss the same topic, pointing out how it broadens the scope, whether they reinforce or oppose the previous findings obtained. The authors should discuss the documentary research in the results section or make it clear that such previous research did not exist or was not found.
3. I suggest including the Future Works section where the authors describe how to improve the limitations that arose during the development of this research and the evaluation of how the results achieved could be effectively adopted.
4. For a better quality of the presentation of results, it is suggested to design one or two graphs that concentrate the most representative results, possibly you can take advantage of the results of appendix A.
Reviewer 4 Report
This manuscript is well written and sounds logical.
Nevertheless, there are opinions to improve the completeness.
1. References need to be updated. I think there will be more recent research.
2. Please describe the detailed inclusion criteria for the participant. (i.e., health profession students?)
3. Some parts do not conform to the submission guidelines. (i.e., references)
4. It is necessary to organize the result table.
Round 2
Reviewer 1 Report
Dear Authors
Thank you for your efforts. Amendments have been made substantially. Acceptable for publication.
Author Response
There were no comments to respond to from R1, I have attached the table of all suggestions for information.

Reviewer 2 Report
Dear authors,
thank you very much for the provided answers. I appreciate a thorough and well-arranged list of responses to my comments.
There is a level of improvement within the provided article. Though, I think that the responses might be utilized more and implemented in the text itself to a larger extent.
At least the socioeconomic context might be worth adding for readers not familiar with the Tasmanian context. As well as the lack of functional digital health literacy assessment and its impacts on the curriculum might be stressed deeply. Especially after the addition of the research questions, which I welcome.
Reviewer 4 Report
1. Please double-check submission guidelines (e.g., Data collection section line 170 -reference style AND I couldn't find a reference on the reference list)
2. In the College of Health and Medicine at the University of Tasmania, several major is included. So I would like to recommend describing more detailed inclusion criteria such as major (i.e., nursing, health science, medicine, pharmacy etc) and also describe at result section.
